# Extraction and Purification of Highly Active Astaxanthin from *Corynebacterium glutamicum* Fermentation Broth

**DOI:** 10.3390/md21100530

**Published:** 2023-10-11

**Authors:** Jan Seeger, Volker F. Wendisch, Nadja A. Henke

**Affiliations:** Genetics of Prokaryotes, CeBiTec, Bielefeld University, 33615 Bielefeld, Germany

**Keywords:** astaxanthin, *Corynebacterium glutamicum*, extraction, antioxidant, DPPH

## Abstract

The marine carotenoid astaxanthin is one of the strongest natural antioxidants and therefore is used in a broad range of applications such as cosmetics or nutraceuticals. To meet the growing market demand, the natural carotenoid producer *Corynebacterium glutamicum* has been engineered to produce astaxanthin by heterologous expression of genes from the marine bacterium *Fulvimarina pelagi*. To exploit this promising source of fermentative and natural astaxanthin, an efficient extraction process using ethanol was established in this study. Appropriate parameters for ethanol extraction were identified by screening ethanol concentration (62.5–97.5% *v*/*v*), temperature (30–70 °C) and biomass-to-solvent ratio (3.8–19.0 mg_CDW_/mL_solvent_). The results demonstrated that the optimal extraction conditions were: 90% ethanol, 60 °C, and a biomass-to-solvent ratio of 5.6 mg_CDW_/mL_solvent_. In total, 94% of the cellular astaxanthin was recovered and the oleoresin obtained contained 9.4 mg/g astaxanthin. With respect to other carotenoids, further purification of the oleoresin by column chromatography resulted in pure astaxanthin (100%, HPLC). In addition, a 2,2-diphenyl-1-picrylhydrazyl (DPPH) radical scavenging assay showed similar activities compared to esterified astaxanthin from microalgae and a nine-fold higher antioxidative activity than synthetic astaxanthin.

## 1. Introduction

Initially used as a feed additive for fish and crustaceans [1,2], the red-colored marine carotenoid astaxanthin has gained much attention for human consumption due to its various health-promoting effects. Based on its molecular structure, consisting of a hydrocarbon backbone with conjugated C-C double bonds (nonpolar) and terminal oxy-functionalized ionone rings (polar) at both sides [3], astaxanthin exhibits anti-inflammatory [4,5], anti-cancer [6] as well as cardioprotective activities [7,8]. As its antioxidative activity is 100 times stronger than α-tocopherol [9], astaxanthin is also used for UV protection and anti-aging applications in the cosmetics industry [10,11]. While still being dominated by astaxanthin obtained by chemical synthesis [12,13], the astaxanthin market is predicted to grow with a CAGR of 17.2%, reaching USD 6.9 billion in 2030 [14]. However, synthetic astaxanthin is not considered for human consumption [15] resulting in an increasing demand for natural astaxanthin. Common hosts for the production of natural astaxanthin are the microalgae *Haematococcus lacustris* (formerly *Haematococcus pluvialis*) (Chlorophyta), the yeast *Xanthophyllomyces dendrorhous* and the Gram-negative bacterium *Paracoccus carotinifaciens* [9,12,16]. Another source of astaxanthin that has been exploited are shells of crustaceans that occur as byproducts in food processing [17,18,19]. 

The costs of downstream processing contribute significantly to the overall production costs. Here, the capital investment and operational costs for each unit operation as well as the overall efficacy to recover the product from the cultivation broth need to be considered [20,21]. Due to its polar-nonpolar-polar structure, astaxanthin is incorporated into the cell membrane [22] or stored intracellularly within lipid droplets [23], depending on the organism. For both, microalgae and yeast, classical preprocessing methods such as (freeze-)drying [16,24,25], ball-milling [13], and high-pressure homogenization [26] have been used prior to extraction. Other processes to disrupt or permeabilize the cell envelope involved enzymatic treatment [27], pulsed electric fields [23], microwaves [28] or ultrasound [27,29]. The extraction itself can be carried out by organic solvents [13,24,25], supercritical fluids [26,30], vegetable oils [31] or eutectic solvents [32]. 

The natural producer of the C50 carotenoid decaprenoxanthin *C. glutamicum* [33], known for amino acid production in million-ton scale [34], has been engineered to produce astaxanthin. Therefore, the β-carotene hydroxylase (CrtZ) and β-carotene ketolase (CrtW) from *Fulvimarina pelagi*, a Mn(II) oxidizing marine bacterium [35], were introduced into the strain [36]. The production was further improved by constructing a fusion protein of CrtZ and CrtW (CrtZ~W), resulting in a promising host for large-scale astaxanthin production (see Appendix A) [37]. Compared to algae-derived astaxanthin, which occurs mainly as mono- and diesters, bacterial astaxanthin is synthesized in an unesterified form [9,12].

As previous studies concerning astaxanthin extraction have shown, several opportunities exist how astaxanthin can be extracted, depending on the used organisms and chemicals. In this study, a fast and simple extraction process using organic solvents should be established, resulting in an astaxanthin oleoresin, which is to be treated further to obtain a purified product. For the choice of extraction solvent, solubility of the product, toxicity and ecological impact are important points to be considered [38]. The solvents, namely ethanol, acetone and ethyl acetate, were chosen from the intersecting set of preferrable solvents for green chemistry [39] and the European guidelines for solvents approved for the production of foodstuffs and food ingredients [40]. Subsequently, the solvent polarity, the extraction temperature and the biomass-to-solvent ratio were optimized, as theses parameters critically effect the extraction efficiency [41]. Finally, the antioxidant activity of the corynebacterial astaxanthin was determined. 

## 2. Results

### 2.1. Optimization of Astaxanthin Extraction Parameters

In the first instance, ethanol, acetone and ethyl acetate were tested for their suitability for the extraction of astaxanthin from *C. glutamicum* cells. Extraction with ethanol yielded 1.4 ± 0.1 mg/g_CDW_ astaxanthin, which was significantly more than the 0.88 ± 0.1 mg/g_CDW_ and 1.1 ± 0.03 mg/g_CDW_ that were extracted by acetone and ethyl acetate, respectively (Figure 1A). Although ethanol seemed to be most promising, the extracted astaxanthin amount corresponded to only 64.7 ± 0.1% of the initial cellular astaxanthin content. Apart from astaxanthin, the precursor carotenoids β-carotene and lycopene were extracted to a non-proportional lesser extent, namely 32.6 ± 0.1% and 27.1 ± 0.1%, resulting in partial depletion of non-targeted carotenoids. To improve the extraction efficiency, the polarity of the solvent was changed by altering the ratio of ethanol and water within the extraction mixture. As shown in Figure 1B, the addition of water improved the extraction efficiency compared to pure ethanol. At 90% ethanol, the extraction efficiency reached its maximum with 98.9 ± 2.3% and remained stable up to 80% ethanol. A further reduction below 80% ethanol resulted in a clearly decreased extraction efficiency. To verify that the extraction temperature of 60 °C from the extraction protocol for corynebacterial carotenoids [42] still applies or could potentially be reduced, temperatures ranging from 30 to 70 °C were tested (Figure 1C). The results showed a clear optimum at 60 °C with lower extraction efficiencies below and above this value. So far, a biomass-to-solvent ration of 3.8 mg_CDW_/mL_solvent_ was used. To reduce the amount of used solvent, the biomass-to-solvent ratio was altered in a range of 3.8–19.0 mg_CDW_/mL_solvent_ (Figure 1D). The cut-off in extraction efficiency was set to 90%, which was reached at a ratio of 5.6 mg_CDW_/mL_solvent_. 

### 2.2. Preparation of Astaxanthin Oleoresin

In order to obtain quantitative amounts of astaxanthin, the optimized extraction parameters were used to scale up the extraction process from a small-scale shaking system (reaction tubes; volume = 1 mL) into a technical scale with a stirred vessel (volume = 1 L). As the agitation of the milliliter system, which was used for the optimization of extraction parameters, could not directly be transferred into the stirred system, the influence of the agitation rate was investigated by a volumetric mass transfer model. The linear correlation between the volumetric mass transfer coefficient of astaxanthin from the biomass into the solvent is shown in Figure 2, where the *k*_L_*a* increased from 0.39 min^−1^ at 200 rpm to 0.56 min^−1^ at 500 rpm. Subsequently, 500 rpm was chosen for further extractions. 

Applying the agitation of 500 rpm determined in the *k*_L_*a* experiment, 2.07 mg/g astaxanthin was extracted into the liquid phase, which can be considered as the complete cellular content (Table 1). As already observed during the solvent screening, some carotenoids, such as β-carotene, lycopene and echinenone were not completely extracted, which contributed to astaxanthin purity. During the removal of the solvent by vacuum rotary evaporation, the solvent was recovered and could potentially be used for further extractions. Overall, the resulting oleoresin contained 9.4 mg/g astaxanthin and 14.7 mg/g total carotenoids (Table 2), corresponding to a recovery of 94% of the initial cellular astaxanthin content. 

### 2.3. Astaxanthin Purification by Column Chromatography

The astaxanthin containing oleoresin, obtained by rotary vacuum evaporation, was loaded onto a C18 column for purification and the collected fractions were analyzed to identify the astaxanthin fraction. HPLC analysis showed that all precursor carotenoids that were present in the oleoresin (Figure 3A) were separated and a high purity astaxanthin fraction was obtained (Figure 3B). In total, 80% of the loaded astaxanthin was collected as pure astaxanthin. Both the astaxanthin oleoresin and the purified astaxanthin were used for subsequent testing of antioxidant activities.

### 2.4. Antioxidant Properties of Corynebacterial Astaxanthin 

A DPPH radical scavenging assay [43] was used to assess the antioxidant properties of the corynebacterial astaxanthin both in the oleoresin and in the column-purified astaxanthin. The astaxanthin purified by column chromatography possessed an EC_50_ value of 4.5 ± 0.2 µg/mL, thus, showing about five-fold higher antioxidant activity as compared to BHT and ascorbic acid and about nine-fold higher antioxidant activity than synthetic astaxanthin (EC_50_ of 41.9 ± 0.7 µg/mL; Table 3). The EC_50_ value of the corynebacterial astaxanthin oleoresin was found to be 3.7 ± 0.6 µg/mL, while the astaxanthin esters from *H. lacustris* had a value of 3.2 ± 0.2 µg/mL. 

## 3. Discussion

In order to meet the growing market demand as well as consumer requirements for natural astaxanthin, the industrial workhorse *C. glutamicum* has been engineered to produce astaxanthin [36,37]. To be economically competitive, not only must the bacterial cultivation be optimized, but also the downstream process for product recovery [20,21]. The recovery of astaxanthin is particularly challenging as the molecule is incorporated within the biomass [22,23], as well as being susceptible to heat, light and oxygen [44]. Since dehydration of the biomass is costly [9], this step should be avoided. Instead, the fermentation broth was centrifuged and the cell pellet was directly used for extraction (Figure 4). Among the three organic solvents ethanol, acetone and ethyl acetate tested for the extraction of astaxanthin, ethanol showed the most promising results. Ethanol has also been used for the extraction of astaxanthin from *X. dendrorhous* [23,45], shrimp [18] and *H. lacustris* [13,25]. To improve the extraction, the polarity of the solvent was varied and showed an optimum with 98.9% extraction efficiency at a solvent-water ratio of 90%. Ahmad and colleagues also found that 90% ethanol worked best for the extraction of astaxanthin from microalgae [46]. The optimum extraction temperature was determined to be at 60 °C. The decrease in extraction efficiency at higher temperatures could be explained by the breakdown of astaxanthin as it was shown that astaxanthin already degrades at 70 °C [47]. In the next optimization step, the biomass-to-solvent ratio was almost doubled, compared to the initial ratio, reaching 5.60 mg_CDW_/mL_solvent_. However, this ratio is still low in relation to other studies, where biomass-to-solvent ratios of 16.7, 181.8, 250 mg_CDW_/mL_solvent_ were applied for *X. dendrorhous* [45], *H. lacustris* [25] and *Jaagichlorella luteoviridis* (formerly *Chlorella luteoviridis*) (Chlorophyta) [46], respectively. The optimized extraction conditions were successfully scaled up into a liter scale to obtain quantitative amounts of corynebacterial astaxanthin. After solvent removal by vacuum rotary evaporation, the astaxanthin oleoresin contained 9.4 mg/g astaxanthin, which is in the same range as oleoresins obtained by ethanol extraction from shrimp (3.4 mg/g [48], 15.6 mg/g [18]). By contrast, the extraction of *H. lacustris* with supercritical carbon dioxide (scCO_2_) yielded oleoresins that contained 96.2 mg/g [49] and 125 mg/g [50] astaxanthin which is about one magnitude higher. A total astaxanthin recovery of 94% was achieved which is higher than the recoveries obtained by Molino et al. [13] using ethanol (67% recovery) and acetone (86% recovery) in an accelerated solvent extraction process. For the extraction using scCO_2_, 80.6% of the cellular astaxanthin content could be recovered [50]. With respect to other carotenoids, column chromatography purification of the corynebacterial astaxanthin oleoresin yielded 100% (HPLC) pure astaxanthin. A purity of 85.1% was achieved by Hu and colleagues [48], who used a silica gel column in comparison to the C18 column used in this study.

To investigate the antioxidant properties of the corynebacterial astaxanthin, the DPPH radical scavenging assay was used. Initially introduced by Marsden Blois in 1958 [43], this assay has been applied in various studies to determine the antioxidant activities of natural compounds and extracts [51,52,53,54]. The assay is based on the neutralization of the DPPH radical by donated electrons from the antioxidants, which results in an absorption shift of DPPH [43]. EC_50_ values of the positive controls ascorbic acid (22.9 µg/mL) and BHT (22.4 µg/mL) were in line with the published data obtained by Chintong et al. [18]. The astaxanthin containing oleoresin of *C. glutamicum* showed a higher antioxidant activity (EC_50_ = 3.7 µg/mL) compared with different extracts from shrimp (17.5 µg/mL; 6.3 µg/mL) [18,55], crab (50.93 µg/mL) [19], *Chromochloris zofingiensis* (formerly *Chlorella zofingiensis*) (Chlorophyta) (1040 µg/mL) [56] and the common astaxanthin production host *X. dendrorhous* (31.79 µg/mL) [45]. We also measured the activity of esterified astaxanthin from *H. lacustris* (3.2 µg/mL), which was comparable to our corynebacterial extract. Recently, another study measured EC_50_ values ranging from 15.39 to 56.25 µg/mL for *H. lacustris* extracts [57]. The purified corynebacterial astaxanthin had an EC_50_ value of 4.5 µg/mL which is more than nine times higher than the activity of the pure synthetic astaxanthin (41.9 µg/mL). Column chromatography purified astaxanthin from *Rhodotorula toruloides* (formerly *Rhodosporidium toruloides*) (Fungi, Basidiomycota) showed an even higher activity with an EC_50_ value of 0.97 µg/mL [24]. The superior activity of the astaxanthin produced by *C. glutamicum* compared to the synthetic astaxanthin might be explained by the molecule’s different stereoisomers. In bacteria and algae, the (3S,3′S) isomer is predominantly produced, while the synthetic version consists of a 1:2:1 mixture of the three isomers (3S,3′S, 3R,3′S and 3R,3′R) [47]. The strain used in this study expresses the β-carotene hydroxylase and β-carotene ketolase from the marine bacterium *Fulvimarina pelagi* [36,37], synthesizing the (3S,3′S) isomer. Using superoxide anion radical as well as hydroxyl radical assays, it was shown that the antioxidant activity of synthetic astaxanthin was inferior compared to natural astaxanthin comprised of the (3S,3′S) isomer [58]. This is in line with the results from Liu and colleagues, who also found that the (3S,3′S) isomer had a higher antioxidant activity in the ABTS radical scavenging assay and a superior oxygen radical absorbance capacity (ORAC) than the other isomers [59]. Interestingly, the latter study found no differences among the three isomers using the DPPH radical scavenging assay. Discrepancies in the DPPH assay parameters influence the experimental outcome [60] and therefore, a direct comparison of the antioxidant activities to our results is difficult. In general, the reason for the higher antioxidant activity of the (3S,3′S) isomer remains to be solved. Compared to the oleoresin, the activity of the purified astaxanthin was slightly lower. This might be due to the additional presence of other carotenoids such as β-carotene and canthaxanthin in the oleoresin, contributing to the antioxidant activity [61,62]. Similar synergistic effects were also observed by Sindhu and Sherief who referred the high antioxidant activity of their shrimp extract to the combination of astaxanthin and poly unsaturated fatty acids measured by different in vitro antioxidant activity assays [63].

**Figure 4 marinedrugs-21-00530-f004:**
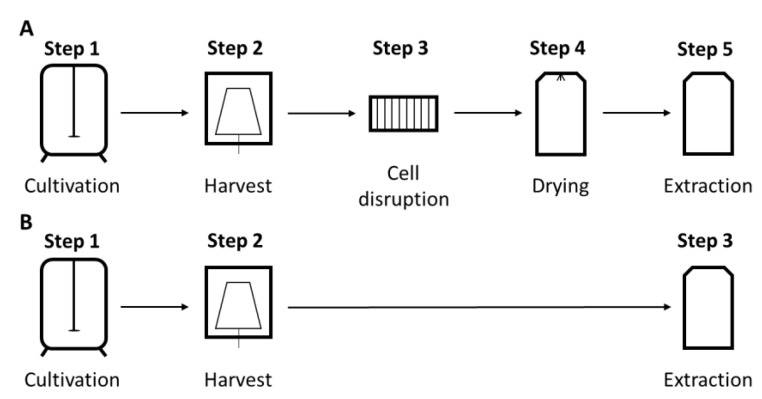
Comparison of astaxanthin downstream processes. Unit operations of each process are shown schematically. Options for further purification or treatment of the extract, e.g., column chromatography, are not shown. (**A**) Common downstream process for algae and yeast derived astaxanthin [64,65]. (**B**) Downstream process of astaxanthin derived from *C. glutamicum* established in this study.

Compared to other astaxanthin downstream processes from algae or yeast [64], this study provides a fast and simple workflow, without the need for extensive equipment or expensive chemicals (Figure 4). In the end, 94% product recovery within the astaxanthin oleoresin was achieved. Both the oleoresin and the purified astaxanthin showed high antioxidative activity in an in vitro DPPH assay. However, further studies are necessary to test the suitability of this corynebacterial astaxanthin for application as a cosmetic ingredient. Therefore, the antioxidant activity could be investigated by cell-based in vitro assays, e.g., with keratinocytes [65] or with stem-cell-based complex skin models [66,67]. Another aspect to be considered is, that astaxanthin, as all carotenoids, is poorly soluble in water and possesses a low bioavailability [11]. Furthermore, free astaxanthin shows a reduced stability compared to the esterified version [68]. These limitations could be overcome by different delivery systems, e.g., liposomes, emulsions and nanoparticles [11,44].

## 4. Materials and Methods

### 4.1. Chemicals and Biomass

If not stated differently, chemicals were purchased by Carl Roth (Karlsruhe, Germany) or Sigma-Aldrich (St. Louis, MA, USA). Solvents for extraction and analysis were HPLC grade. The astaxanthin producing *C. glutamicum* strain (Pathway: Appendix A) was cultivated in CGXII minimal medium, supplemented with 4% (*w*/*v*) glucose in baffled shake flasks on a rotary shaker (120 rpm) at 30 °C [37]. For the optimization of the extraction parameters, 0.5 mL culture was harvested after 48 h of cultivation at 20,000× *g* for 10 min. For the extraction in the stirred bottle reactor, culture was harvested accordingly after 48 h of cultivation at 10,000× *g* for 20 min. Water content of the cell pellet was determined by drying it completely.

### 4.2. Optimization of Extraction Parameters

#### 4.2.1. Solvent Screening

In order to find an appropriate solvent for the extraction of astaxanthin, ethanol, acetone and ethyl acetate were tested. The cell pellet was extracted with 1 mL solvent at 1000 rpm for 30 min (Thermomixer comfort, Eppendorf, Hamburg, Germany). The biomass-to-solvent ratio and temperature were kept constant at 3.8 mg_CDW_/mL_solvent_ and at 60 °C, respectively. After centrifugation at 20,000× *g* for 10 min, the supernatant was analyzed via HPLC. The extraction efficiency was determined by comparison with the extraction protocol for corynebacterial carotenoids used for analytical purposes (7:3 mixture of methanol:acetone), which assumes that all carotenoids are extracted from the cell [42]. The extraction was performed in triplicates.

#### 4.2.2. Ethanol Concentration

To determine the optimal ethanol concentration, the water content and the dry mass of harvested cultures were determined. Based on the water content of the biomass, absolute ethanol and ddH_2_O were added accordingly, to reach the desired ethanol concentration (*v*/*v*). As the cell pellet already contained a certain amount of water, ethanol concentrations ranging from 62.5 to 97.5% (*v*/*v*) were tested. The cell pellet was extracted with 1 mL total solvent at 1000 rpm for 30 min (Thermomixer comfort, Eppendorf, Hamburg, Germany). The biomass-to-solvent ratio and temperature were kept constant at 3.8 mg_CDW_/mL_solvent_ and at 60 °C, respectively. The extraction efficiency was determined by comparison with the extraction protocol for corynebacterial carotenoids (7:3 mixture of methanol:acetone) [42]. The extraction was performed at least in duplicates.

#### 4.2.3. Temperature

To determine the optimal extraction temperature, samples were extracted with 1 mL 90% (*v*/*v*) ethanol at temperatures ranging from 30 to 70 °C, following the same procedure as in Section 4.2.1. The extraction efficiency was determined by comparison with the extraction using 90% (*v*/*v*) ethanol at 60 °C (optimum from Section 4.2.2). The extraction was performed in triplicates.

#### 4.2.4. Biomass-to-Solvent Ratio

Under previous conditions, 3.8 mg_CDW_/mL_solvent_ was used for extraction. To determine the optimal biomass-to-solvent ratio, the already optimized parameters were used and biomass-to-solvent ratios ranging from 3.8 to 19.0 mg_CDW_/mL_solvent_ were tested. The extraction efficiency was determined by comparison with the biomass-to-solvent ratio of 3.8 mg_CDW_/mL_solvent_ (optimum from Section 4.2.3). The extraction was performed in triplicates.

### 4.3. Astaxanthin Extraction

Astaxanthin extraction was performed in a 1 L stirred bottle reactor equipped with an anchor stirrer (DWK Life Sciences, Mainz, Germany). Temperature and agitation were controlled by a magnetic stirrer with heating plate and temperature probe (IDL GmbH, Nidderau, Germany). Centrifuged biomass from cultivation with a water content of 80–85% (*w*/*w*) was used. Based on the amount of dry substance, respective amounts of absolute ethanol and ddH_2_O were added to reach 5.6 mg_CDW_/mL_solvent_ with an ethanol concentration of 90% (*v*/*v*). Extraction was performed at 60 °C and 500 rpm for 20 min. Liquid crude extract was analyzed by HPLC and was used for preparation of the astaxanthin oleoresin. The extraction efficiency was determined by comparison with the extraction protocol for corynebacterial carotenoids (7:3 mixture of methanol:acetone) [42].

#### Kinetic Model

To describe the extraction kinetic and to evaluate the influence of different agitation rates on the extraction, the mass transfer kinetic model proposed by Handayani et al. [69] was applied. This model assumes that the limiting step of the extraction is the mass transfer of astaxanthin from the biomass into the solvent. The rate of mass transfer can be written as:d*N*_A_/d*t* = *k*_L_ ∗ *A* ∗ [*C*_Ae_ − *C*_A_](1)
with d*N*_A_/d*t* as the rate of astaxanthin mass transfer [mg/min], *C*_A_ and *C*_AE_ are the concentrations of astaxanthin in liquid and at equilibrium [mg/L], respectively. *k*_L_ is the mass transfer coefficient and *A* the surface area. As the process was carried out in batch mode, the volume (*V*) was kept constant.
d*N*_A_ = *V*d*C*_A_(2)

Substitution of (2) into (1) results in
*V*d*C*_A_/d*t* = *k*_L_ ∗ *A * ∗ [*C*_Ae_ − *C*_A_](3)
d*C*_A_/d*t* = *k*_L_ ∗ *A*/*V * ∗ [*C*_Ae_ − *C*_A_](4)
d*C*_A_/d*t = k*_L_ ∗ *a* ∗ [*C*_Ae_ − *C*_A_](5)

With *k*_L_ ∗ *a* being the volumetric mass transfer coefficient. Considering that at the beginning of the process (*t* = 0), the astaxanthin concentration in the liquid is zero (*C*_A_ = 0) and the concentration of astaxanthin at any time is *C*_A_ = *C*_A_, integration of (3) yields
*C*_A_ = *C*_AE_ ∗ [1 − exp (−*k*_L_ ∗ *a ∗ t*)](6)

For this model, the parameters *C*_A_, *k*_L_ and *a* were estimated by nonlinear least squares fit of the experimental data. The agitation rate was varied as indicated and samples were drawn at 2, 5, 10, 15, 20, 25, 30, 35, 40 min.

### 4.4. Preparation of Astaxanthin Oleoresin

Liquid crude extract (Section 4.3) was concentrated to one-twentieth of the initial volume by vacuum rotary evaporation (VV2000, Heidolph Instruments, Schwabach, Germany). Absolute ethanol was added in a relation of 7:1 to the concentrated crude extract and the mixture was vigorously shaken for 3 min. Supernatant was separated from solid precipitate and the liquid phase was removed by vacuum rotary evaporation. The oleoresin obtained was stored at −20 °C until further usage. The carotenoid recovery was calculated by comparison with the cellular carotenoid content determined with the extraction protocol for corynebacterial carotenoids (7:3 mixture of methanol:acetone) [42].

### 4.5. Purification by Column Chromatography

For further purification, astaxanthin oleoresin was resolubilized in methanol and injected into a flash chromatography system (Reveleris X2, Büchi Labortechnik, Flawil, Switzerland) equipped with a 12 g FlashPure EcoFlex C18 column (Büchi Labortechnik, Flawil, Switzerland). Methanol:water (9:1) (A) and methanol (B) were used as mobile phases. The injection volume was 1 mL and a gradient flow at a rate of 30 mL min^−1^ was used as per the following: 0 min B: 0%, 8 min B: 100%, 25.7 min B: 100%. Collected fractions were analyzed by HPLC.

### 4.6. Quantification of Carotenoids

The quantification of carotenoids (Structure: Appendix A) was performed as previously described [42]. Standards were used for standard calibration curves using lycopene (ExtraSynthese, Genay, France), β-carotene (Sigma-Aldrich, St. Louis, MA, USA), canthaxanthin (VWR, Darmstadt, Germany), echinenone (Sigma-Aldrich, St. Louis, MA, USA), adonirubin (CaroteNature, Münsingen, Switzerland), 3-hydroxyechinenone (CaroteNature, Münsingen, Switzerland) and astaxanthin (Sigma-Aldrich, St. Louis, MA, USA).

### 4.7. DPPH Assay

The radical scavenging activity test was carried out as described [18] with slight modifications. Serial dilutions of the respective antioxidant (3.125–100 µg/mL; for column chromatography purified astaxanthin: 0.64–20.5 µg/mL) were prepared in methanol. In case of the oleoresin from *C. glutamicum* and the esterified astaxanthin from *H. lacustris*, the antioxidant concentration refers to the amount of free astaxanthin. In total, 0.4 mL of each dilution was mixed with 0.4 mL of the DPPH solution (0.18 mM in methanol) and incubated at room temperature for 30 min in the dark. The absorbance at 517 nm was determined photometrically (UV-VIS Spectrophotometer UV-1650PC, Shimadzu, Kyoto, Japan). The radical scavenging activity was calculated as follows:DPPH scavenging activity (%) = (A_control_ − (A_sample_ − A_sample blank_))/A_control_(7)
with A_control_, A_sample_ and A_sample blank_ being the absorbances of the DPPH solution without antioxidant, antioxidant solution with DPPH and the antioxidant solution without DPPH, respectively. EC_50_ value (efficient concentration when 50% of the radial is reduced) was expressed in µg/mL (related to the respective antioxidant) and was calculated by plotting the antioxidant concentration against the scavenging activity. The measurements were performed in triplicates. Butylated hydroxytoluene (Merck KGaA, Darmstadt, Germany), ascorbic acid (Karlsruhe, Germany), astaxanthin esters from *Haematococcus pluvialis* (Sigma-Aldrich, St. Louis, MA, USA) and synthetic astaxanthin (Sigma-Aldrich, St. Louis, MA, USA) were used for comparison.

## Figures and Tables

**Figure 1 marinedrugs-21-00530-f001:**
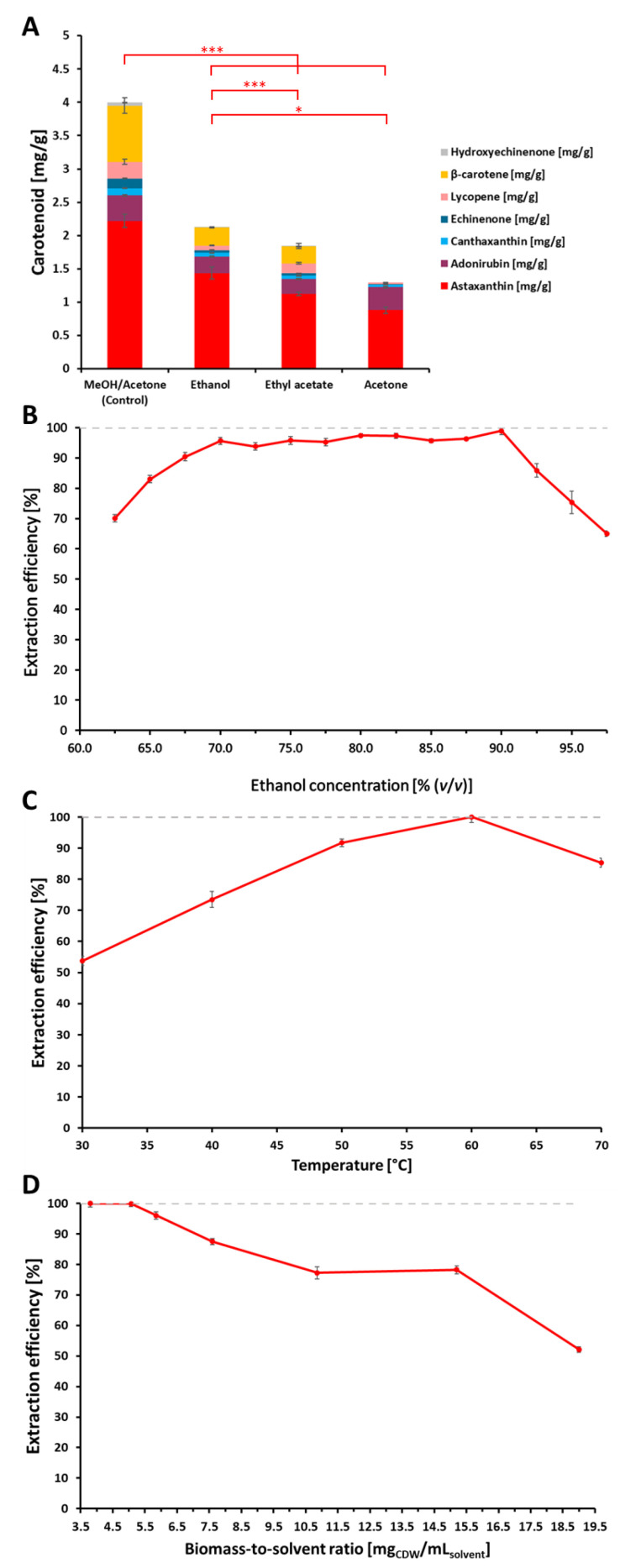
Optimization of astaxanthin extraction parameters. (**A**) Solvent screening. Extraction yield is given in mg/g for methanol:acetone (7:3) as control, ethanol, acetone and ethyl acetate. Statistical differences are given for astaxanthin (* for *p* < 0.05, *** for *p* < 0.001). (**B**) Ethanol concentration. Extraction efficiency [%] of astaxanthin is given in relation to methanol:acetone (7:3) as control. (**C**) Extraction temperature. Extraction efficiency [%] of astaxanthin is given in relation to the extraction at 90% ethanol at 60 °C. (**D**) Biomass-to-solvent ratio. Extraction efficiency [%] of astaxanthin is given in relation to the extraction at 90% ethanol at 60 °C and 3.8 mg_CDW_/mL_solvent_.

**Figure 2 marinedrugs-21-00530-f002:**
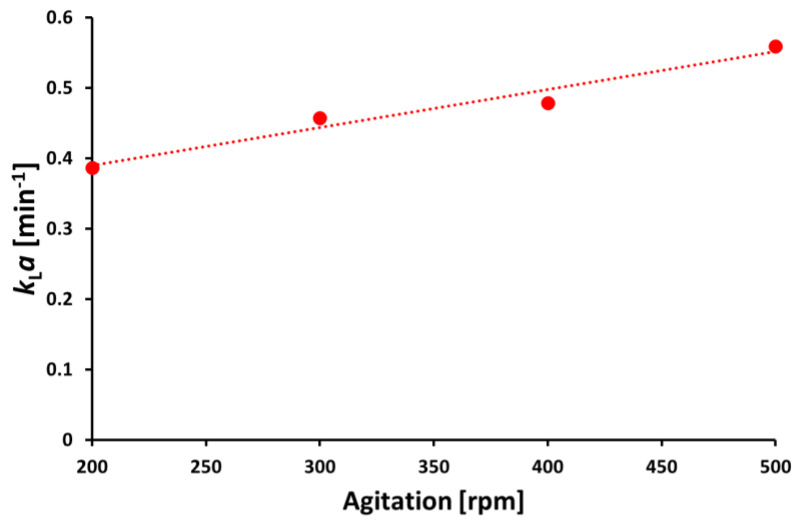
Volumetric mass transfer coefficient of astaxanthin. k_L_a values in stirred extraction vessel at 60 °C with varying agitation rates calculated using mass transfer kinetic model. R^2^ = 0.96.

**Figure 3 marinedrugs-21-00530-f003:**
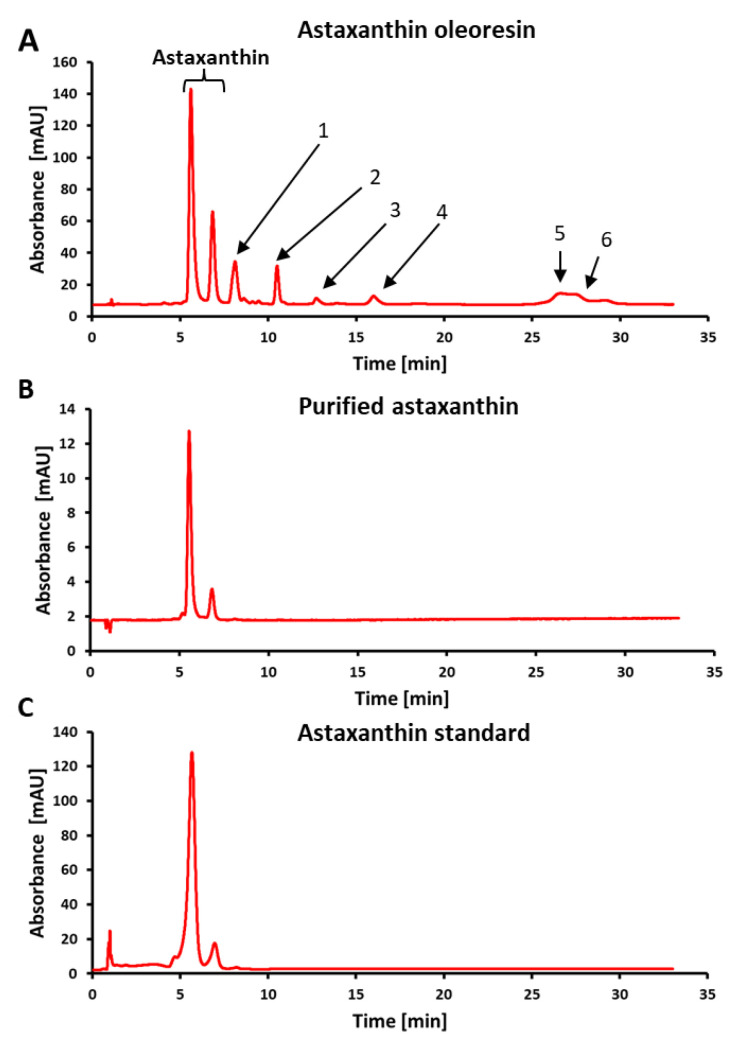
Purification of astaxanthin by column chromatography. (**A**) Astaxanthin oleoresin from *C. glutamicum*. 1: Adonirubin, 2: Canthaxanthin, 3: Hydroxyechinenone, 4: Echinenone, 5: Lycopene, 6: β-carotene. (**B**) Purified astaxanthin from oleoresin. (**C**) Astaxanthin standard.

**Table 1 marinedrugs-21-00530-t001:** Astaxanthin extraction in stirred vessel. Extraction yield of the respective carotenoid is given in mg/g_CDW_. Extraction efficiency [%] is given in relation to the initial cellular carotenoid content determined by the control extraction.

Carotenoid	Extraction Yield [mg/g_CDW_]	Extraction Efficiency [%]
Astaxanthin	2.07	108
Adonirubin	0.30	101
Canthaxanthin	0.10	94
Echinenone	0.02	20
Hydroxyechinenone	0.03	106
Lycopene	0.07	11
β-carotene	0.41	67
Total carotenoids	3.00	91

**Table 2 marinedrugs-21-00530-t002:** Carotenoid content in astaxanthin oleoresin. Content is given in mg/g_oleoresin_. Recovery [%] is given in relation to the initial cellular carotenoid content determined by the control extraction.

Carotenoid	Oleoresin [mg/g_oleoresin_]	Recovery [%]
Astaxanthin	9.41	94
Adonirubin	1.75	112
Canthaxanthin	0.48	87
Echinenone	0.11	18
Hydroxyechinenone	0.13	94
Lycopene	0.48	14
β-carotene	2.37	70
Total carotenoids	14.7	75

**Table 3 marinedrugs-21-00530-t003:** Antioxidant properties of corynebacterial astaxanthin. Antioxidant activities are given as EC_50_ [µg/mL] in comparison to other antioxidants.

Antioxidant	EC_50_ [µg/mL]
BHT	22.4 ± 0.5 ^a^
Ascorbic acid	22.9 ± 0.1 ^a^
Synthetic astaxanthin	41.9 ± 0.7
Astaxanthin esters from *H. lacustris*	3.2 ± 0.2 ^b^
Astaxanthin oleoresin from *C. glutamicum*	3.7 ± 0.6 ^b^
Purified astaxanthin from *C. glutamicum*	4.5 ± 0.2 ^b^

^a,b^ Mean values followed by the same letter are not significantly different to each other.

## Data Availability

Not applicable.

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
