# Peer review of "Extraction and Purification of Highly Active Astaxanthin from Corynebacterium glutamicum Fermentation Broth"

_marinedrugs, 2023, doi:10.3390/md21100530_

Round 1

Reviewer 1 Report

In this study, Corynebacterium glutamicum has been engineered to produce astaxanthin by heterologous expression of genes from the marine bacterium Fulvimarina pelagi. The research obtained the optimal extraction conditions were: 90% ethanol, 60 °C, and a biomass-to-solvent ratio of 5.6 mgCDW/mLsolvent. Moreover, 94% of the cellular astaxanthin was recovered and the oleoresin obtained contained 9.4 mg/g astaxanthin. Importantly, 2,2-diphenyl-1-picryredrazyl radical scavenging assay showed similar activities compared to esterified astaxanthin from microalgae and a 9-fold higher antioxidative activity than synthetic astaxanthin. This knowledge provides a good foundation to produce astaxanthin from engineered Corynebacterium glutamicum in the future.

Overall, the manuscript ideas appear justified. I find the paper quite good.

Listed are some comments regarding the submitted manuscript?

Line 94: Please check all the Bold form of legend in the x-axis and y-axis of Figure 1.

Please check all the abbreviation form of strain name (Corynebacterium glutamicum           C. glutamicum) in the text: line 50, 241,…

Author Response

Reviewer 1:

In this study, Corynebacterium glutamicum has been engineered to produce astaxanthin by heterologous expression of genes from the marine bacterium Fulvimarina pelagi. The research obtained the optimal extraction conditions were: 90% ethanol, 60 °C, and a biomass-to-solvent ratio of 5.6 mgCDW/mLsolvent. Moreover, 94% of the cellular astaxanthin was recovered and the oleoresin obtained contained 9.4 mg/g astaxanthin. Importantly, 2,2-diphenyl-1-picryredrazyl radical scavenging assay showed similar activities compared to esterified astaxanthin from microalgae and a 9-fold higher antioxidative activity than synthetic astaxanthin. This knowledge provides a good foundation to produce astaxanthin from engineered Corynebacterium glutamicum in the future.

Overall, the manuscript ideas appear justified. I find the paper quite good. 

Listed are some comments regarding the submitted manuscript?

Line 94: Please check all the Bold form of legend in the x-axis and y-axis of Figure 1.

-_-> Dear reviewer, thanks for your comment. We double checked the bold form of the axis legends and all legends are written in bold.

Please check all the abbreviation form of strain name (Corynebacterium glutamicum →            C. glutamicum) in the text: line 50, 241,…

-_-> As recommended, we changed the abbreviations accordingly.

Reviewer 2 Report

This study investigated the efficient extraction method and DPPH radical scavenging activity of astaxanthin produced by the aerobic Gram-positive bacterium Corynebacterium glutamicum.

There are some things that seem unclear, so I comment on them.

1, Figure 3

There are two peaks for astaxanthin in the figure, but are they all-trans (main peak on the left) and cis-form (peak on the right)? Please indicate in the figure.

 In A, the ratio of these two peaks appears to be about 2:1, but in B and C, it appears to be about 6 to 7:1. Why is this difference?

 In the figure (B) of purified astaxanthin, the scale of the vertical axis is 10 times smaller than in A and C. Even if you increase this scale by 10 times, won't other carotenoid peaks appear?

2, Extraction solvent and purification solvent

The introduction states that “For the choice of extraction solvent, solubility of the product, toxicity and ecological impact are important points to be considered [38]. The solvents, namely ethanol, acetone and ethyl acetate were chosen from the intersecting set of preferrable solvents for green chemistry [39] and the European guidelines for solvents approved for the production of foodstuffs and food ingredients [40]."

 If so, why did the authors use methanol (rather than ethanol) in the purification procedure from astaxanthin oleoresin?

3, DPPH

Although natural astaxanthin has higher antioxidant activity than organically synthesized astaxanthin, there are some concerns.

This is the OH group in the structure of astaxanthin that is involved in scavenging DPPH radicals, and its scavenging ability may depend on the position of its optical activity.

Indeed, the reference 59 cited by the author states that in a comparison of synthetic and natural astaxanthin, the natural one was 50 times more active in singlet oxygen scavenging and 20 times more active in free radical scavenging than the synthetic one. be.

However, in reference 60, although there is a relationship between intracellular ROS and nematode lifespan, there is not much difference in nematode lifespan when comparing optical isomers.

Reference 63 does not consider DPPH radical scavenging ability. Since it is a TBARS and NBT assay, I feel that quoting Reference 63 is inappropriate.

In reference 59, the scavenging ability of natural substances is 20-50 times higher than that of synthetic substances, and in this study, it is 9 times higher. Does the difference in the position of the OH group make such a difference? I feel uncomfortable with this.

Could this result be due to the fact that natural astaxanthin fraction contains some other antioxidant component (with DPPH radical scavenging ability) such as tocopherol, rather than the position of the OH group?

4, Corynebacterium glutamicum

The route by which the strain was obtained is unknown.

Author Response

Reviewer 2:

This study investigated the efficient extraction method and DPPH radical scavenging activity of astaxanthin produced by the aerobic Gram-positive bacterium Corynebacterium glutamicum.

There are some things that seem unclear, so I comment on them.

1, Figure 3

There are two peaks for astaxanthin in the figure, but are they all-trans (main peak on the left) and cis-form (peak on the right)? Please indicate in the figure.

-_-> We observe this peak pattern for both natural astaxanthin (Sigma Aldrich, SML0982, purity according to CoA: >99.5%) and for synthetic astaxanthin. It is not stated by the manufacturer, which isomers are contained in the natural astaxanthin (shown in figure 3C). That is why we did not indicate that in the figure. Indeed, it was shown by other researchers using reversed-phase HPLC that trans-astaxanthin elutes before 13-cis and 9-cis, respectively (https://doi.org/10.1186/s40643-021-00410-5)

 In A, the ratio of these two peaks appears to be about 2:1, but in B and C, it appears to be about 6 to 7:1. Why is this difference?

-_-> Dear reviewer, thanks for your comment. The decrease of the second peak in 3B is based on the purification process during the flash chromatography.

 In the figure (B) of purified astaxanthin, the scale of the vertical axis is 10 times smaller than in A and C. Even if you increase this scale by 10 times, won't other carotenoid peaks appear?

-_-> As recommended, we double checked the absence of other carotenoid peaks in the purified astaxanthin fraction. If we would increase the y-axis scale 10x (same range as for figure 3A and C), the astaxanthin peak in 3B would get very small. Figures 3A/C show a highly concentrated oleoresin and astaxanthin standard, respectively. However, due to the purification by the flash chromatography, the astaxanthin gets diluted, resulting in a low overall peak intensity of the HPLC signal.

2, Extraction solvent and purification solvent

The introduction states that “For the choice of extraction solvent, solubility of the product, toxicity and ecological impact are important points to be considered [38]. The solvents, namely ethanol, acetone and ethyl acetate were chosen from the intersecting set of preferrable solvents for green chemistry [39] and the European guidelines for solvents approved for the production of foodstuffs and food ingredients [40]."

 If so, why did the authors use methanol (rather than ethanol) in the purification procedure from astaxanthin oleoresin?

-_-> Dear reviewer, thanks for your comment. As already mentioned in the introduction, only the extraction solvent was chosen based on the selected guidelines. The oleoresin itself, which is widely used as a starting material in the cosmetics industry, was obtained by only using ethanol.

For the flash chromatography purification however, we used our cited HPLC method for carotenoid analysis and transferred this method to the flash chromatography system. The purification was performed to be able to compare the antioxidant activities of the highly pure astaxanthin from C. glutamicum with the pure synthetic astaxanthin. That is why it was not relevant which solvent was used for the purification by flash chromatography.

The reviewer is right, the purified astaxanthin was not obtained by green/food-approves solvents.   

3, DPPH

Although natural astaxanthin has higher antioxidant activity than organically synthesized astaxanthin, there are some concerns.

This is the OH group in the structure of astaxanthin that is involved in scavenging DPPH radicals, and its scavenging ability may depend on the position of its optical activity.

Indeed, the reference 59 cited by the author states that in a comparison of synthetic and natural astaxanthin, the natural one was 50 times more active in singlet oxygen scavenging and 20 times more active in free radical scavenging than the synthetic one. be.

However, in reference 60, although there is a relationship between intracellular ROS and nematode lifespan, there is not much difference in nematode lifespan when comparing optical isomers.

-_-> Dear reviewer, thanks for your comment. Indeed reference 60 chapter 3.2 showed that there were significant differences in intracellular antioxidant activity between the three isomers with the S-form exhibiting the highest activity. Chapter 3.3 of reference 60 states that there were differences (although not significant) between the three isomers on the lifespan of the nematode: “There was greatest effect by S AST on extending lifespan of C. elegans, followed by R AST and M AST. These findings were in accordance with in vitro chemical antioxidant activity and CAA assay.”  We cannot further comment on the influence of different isomers on the lifespan of nematodes.

Reference 63 does not consider DPPH radical scavenging ability. Since it is a TBARS and NBT assay, I feel that quoting Reference 63 is inappropriate.

-_-> Dear reviewer, thanks for this comment. Indeed, reference 63 used TBARS and NBT assay and not the DPPH assay. However, the intention here was not to compare the antioxidant activities between the astaxanthin from reference 63 to our values. In fact, we wanted to show that also other researchers found that there might be synergistic antioxidant activities of several molecules within a natural extract that contribute to the overall activity. As our results showed, the activity of the oleoresin was higher than the activity of the purified astaxanthin, which could be explained by synergistic effects due to the presence of other carotenoids within the oleoresin. We adjusted our formulation within the discussion.

In reference 59, the scavenging ability of natural substances is 20-50 times higher than that of synthetic substances, and in this study, it is 9 times higher. Does the difference in the position of the OH group make such a difference? I feel uncomfortable with this.

-_-> Dear reviewer, thanks for this comment. Also referring to you comments above: we reformulated the paragraph within the discussion. We did not to intend to directly compare our results to the results of reference 59. However, we wanted to compare the trends in antioxidant activity which are similar to our findings.

In reference 60, also three different in vitro chemical antioxidant tests were performed: DPPH, ABTS and ORAC. The (3S,3´S) isomer showed the highest activity for ABTS and ORAC. Again, the observed trends in activity match with our findings. Interestingly, the activities for DPPH showed no differences among the three isomers. Important to note is that reference 60 used different parameters for their assay (e.g. different solvent and 4x longer incubation). It is known that especially time is a critical parameter and should be kept constant for a best comparison. That is why it would be interesting to see the results for the three different isomers with our conditions. During the selection of antioxidant activities of astaxanthin from different sources, we paid attention to select studies which uses conditions that are as close as possible to our conditions. Especially the assay time was the same across all studies we are comparing with (30min).  

We adjusted our formulation in the discussion.

Reference 60 does give some suggestions about the reason of the different activities of the three different isomers however, they are not proven yet.

Could this result be due to the fact that natural astaxanthin fraction contains some other antioxidant component (with DPPH radical scavenging ability) such as tocopherol, rather than the position of the OH group?

-_-> Dear reviewer, thanks for this comment. Yes, natural extracts can potentially contain other components that also possess antioxidant activity, as it was shown for our oleoresin. However, it is unlikely that an unknown component of such a strong antioxidant activity is present in the purified fraction of our astaxanthin, that would increase the activity by factor 9 (compared to synthetic astaxanthin).

4, Corynebacterium glutamicum

The route by which the strain was obtained is unknown.

-_-> As recommended, we described the cultivation of the strain in the methods section (see chapter 4.1) more detailed. In addition, we added a schematic biosynthetic route of astaxanthin to the supplementary material. 

Reviewer 3 Report

The manuscript entitled "Extraction and purification of highly active astaxanthin from Corynebacterium glutamicum fermentation broth" addresses a relevant and appropriate topic for this journal. 

The manuscript is well written and well-founded, at the end of the day the authors will have to correct the taxonomic part of the organisms mentioned, because for the manuscript to be rigorous in scientific terms, the authors will have to always use valid names and not synonyms.

Corrections needed: 

line 36 - Haematococcus lacustris (formerly Haematococcus pluvialis) (Chlorophyta), the yeast Xanthophyllomyces dendrorhous and the Gram-negative

line 110 - solvent is shown in Figure 2, where the kLa increased from 0.39 min-1 at 200 rpm to 

line 117 - cellular content (Table 1). As already observed during the solvent screening, some carote- 

line 121 - 9.4 mg/g astaxanthin and 14.7 mg/g total carotenoids (Table 2), corresponding to a recov- 

line 127 - Table 2. Composition of astaxanthin oleoresin. Carotenoid content is given in mg/goleoresin. Recover (Note: "Table 2" is not in italics)

line 133 - that were present in the oleoresin (Figure 3A) were separated and a high purity astaxan-

line 149 - the astaxanthin esters from H. lacustris had a value of 3.2 ± 0.2 µg/mL.

Table 3 - ... Astaxanthin esters from H. lacustris

line 164/165 - ... and H. lacustris [13,25]. To improve the extraction, the polarity of the solvent was varied and showed 

line 174 - plied for X. dendrorhous [45], H. pluvialis [25] and Jaagichlorella luteoviridis (formerly Chlorella luteoviridis) (Chlorophyta) [46], respectively. The

line 179 - 15.6 mg/g [18]). In contrast to that, extraction of H. lacustris with supercritical carbon di-

line 198 - [18,55], crab (50.93 µg/mL) [19], Chromochloris zofingiensis (formerly Chlorella zofingiensis) (Chlorophyta) (1040 µg/mL) [56] and the common 

line 200 - tivity of esterified astaxanthin from H. lacustris (3.2 µg/mL), which was comparable to our

line 202 - 15.39-56.25 µg/mL for H. lacustris extracts [57]. The purified corynebacterial astaxanthin

line 205 - Rhodotorula toruloides (formerly Rhodosporidium toruloides) (Fungi, Basidiomycota) showed an even higher activity with an EC50 value of 0.97 µg/mL

line 336 - the oleoresin from C. glutamicum and the esterified astaxanthin from H. lacustris, the anti-

Author Response

Reviewer 3:

The manuscript entitled "Extraction and purification of highly active astaxanthin from Corynebacterium glutamicum fermentation broth" addresses a relevant and appropriate topic for this journal. 

The manuscript is well written and well-founded, at the end of the day the authors will have to correct the taxonomic part of the organisms mentioned, because for the manuscript to be rigorous in scientific terms, the authors will have to always use valid names and not synonyms.

Corrections needed: 

line 36 - Haematococcus lacustris (formerly Haematococcus pluvialis) (Chlorophyta), the yeast Xanthophyllomyces dendrorhous and the Gram-negative

-_-> Dear reviewer, thanks for this comment. We corrected the name accordingly.

line 110 - solvent is shown in Figure 2, where the kLa increased from 0.39 min-1 at 200 rpm to 

-_-> Dear reviewer, thanks for this comment. We changed “figure” into “Figure”.

line 117 - cellular content (Table 1). As already observed during the solvent screening, some carote- 

-_-> Dear reviewer, thanks for this comment. We changed “table” into “Table”.

line 121 - 9.4 mg/g astaxanthin and 14.7 mg/g total carotenoids (Table 2), corresponding to a recov- 

-_-> corrected

line 127 - Table 2. Composition of astaxanthin oleoresin. Carotenoid content is given in mg/goleoresin. Recover (Note: "Table 2" is not in italics)

-_-> Dear reviewer, thanks for this comment. We changed it to non-italics.

line 133 - that were present in the oleoresin (Figure 3A) were separated and a high purity astaxan-

-_-> corrected

line 149 - the astaxanthin esters from H. lacustris had a value of 3.2 ± 0.2 µg/mL.

-_-> corrected.

Table 3 - ... Astaxanthin esters from H. lacustris

-_-> corrected.

line 164/165 - ... and H. lacustris [13,25]. To improve the extraction, the polarity of the solvent was varied and showed 

-_-> corrected.

line 174 - plied for X. dendrorhous [45], H. pluvialis [25] and Jaagichlorella luteoviridis (formerly Chlorella luteoviridis) (Chlorophyta) [46], respectively. The

-_-> corrected.

line 179 - 15.6 mg/g [18]). In contrast to that, extraction of H. lacustris with supercritical carbon di-

 -_-> corrected.

line 198 - [18,55], crab (50.93 µg/mL) [19], Chromochloris zofingiensis (formerly Chlorella zofingiensis) (Chlorophyta) (1040 µg/mL) [56] and the common 

 -_-> corrected.

line 200 - tivity of esterified astaxanthin from H. lacustris (3.2 µg/mL), which was comparable to our

 -_-> corrected.

line 202 - 15.39-56.25 µg/mL for H. lacustris extracts [57]. The purified corynebacterial astaxanthin

 -_-> corrected.

line 205 - Rhodotorula toruloides (formerly Rhodosporidium toruloides) (Fungi, Basidiomycota) showed an even higher activity with an EC50 value of 0.97 µg/mL

 -_-> corrected.

line 336 - the oleoresin from C. glutamicum and the esterified astaxanthin from H. lacustris, the anti-

-_-> corrected.